# Pea-KD: Parameter-efficient and Accurate Knowledge Distillation on BERT

## Abstract

How can we efficiently compress a model while maintaining its performance? Knowledge Distillation (KD) is one of the widely known methods for model compression. In essence, KD trains a smaller student model based on a larger teacher model and tries to retain the teacher model's level of performance as much as possible. However, existing KD methods suffer from the following limitations. First, since the student model is smaller in absolute size, it inherently lacks model capacity. Second, the absence of an initial guide for the student model makes it difficult for the student to imitate the teacher model to its fullest. Conventional KD methods yield low performance due to these limitations.

In this paper, we propose Pea-KD (Parameter-efficient and accurate Knowledge Distillation), a novel approach to KD. Pea-KD consists of two main parts: Shuffled Parameter Sharing (SPS) and Pretraining with Teacher's Predictions (PTP). Using this combination, we are capable of alleviating the KD's limitations. SPS is a new parameter sharing method that increases the student model capacity. PTP is a KD-specialized initialization method, which can act as a good initial guide for the student. When combined, this method yields a significant increase in student model's performance. Experiments conducted on BERT with different datasets and tasks show that the proposed approach improves the student model's performance by 4.4% on average in four GLUE tasks, outperforming existing KD baselines by significant margins.

## 1 Introduction

*How can we improve the accuracy of knowledge distillation (KD) with smaller number of parameters?* KD uses a well-trained large teacher model to train a smaller student model. Conventional KD method (Hinton et al. (2006)) trains the student model using the teacher model's predictions as targets. That is, the student model uses not only the true labels (hard distribution) but also the teacher model's predictions (soft distribution) as targets. Since better KD accuracy is directly linked to better model compression, improving KD accuracy is valuable and crucial.

Naturally, there have been many studies and attempts to improve the accuracy of KD. Sun et al. (2019) introduced Patient KD, which utilizes not only the teacher model's final output but also the intermediate outputs generated from the teacher's layers. Jiao et al. (2019) applied additional KD in the pretraining step of the student model. However, existing KD methods share the limitation of students having lower model capacity compared to their teacher models, due to their smaller size. In addition, there is no proper initialization guide established for the student models, which especially becomes important when the student model is small. These limitations lower than desired levels of student model accuracy.

In this paper, we propose Pea-KD (Parameter-efficient and accurate Knowledge Distillation), a novel KD method designed especially for Transformer-based models (Vaswani et al. (2017)) that significantly improves the student model's accuracy. Pea-KD is composed of two modules, Shuffled Parameter Sharing (SPS) and Pretraining with Teacher's Predictions (PTP). Pea-KD is based on the following two main ideas.

1. We apply SPS in order to increase the effective model capacity of the student model without increasing the number of parameters. SPS has two steps: 1) stacking layers that share

parameters and 2) shuffling the parameters between shared pairs of layers. Doing so increases the model's effective capacity which enables the student to better replicate the teacher model (details in Section 3.2).

2. We apply a pretraining task called PTP for the student. Through PTP, the student model learns general knowledge about the teacher and the task. With this additional pretraining, the student more efficiently acquires and utilizes the teacher's knowledge during the actual KD process (details in Section 3.3).

Throughout the paper we use Pea-KD applied on BERT model (PeaBERT) as an example to investigate our proposed approach. We summarize our main contributions as follows:

- **Novel framework for KD.** We propose SPS and PTP, a parameter sharing method and a KD-specialized initialization method. These methods serve as a new framework for KD to significantly improve accuracy.
- **Performance.** When tested on four widely used GLUE tasks, PeaBERT improves student's accuracy by 4.4% on average and up to 14.8% maximum when compared to the original BERT model. PeaBERT also outperforms the existing state-of-the-art KD baselines by 3.5% on average.
- **Generality.** Our proposed method Pea-KD can be applied to any transformer-based models and classification tasks with small modifications.

Then we conclude in Section 5.

## 2 RELATED WORK

**Pretrained Language Models.**   The framework of first pretraining language models and then fine-tuning for downstream tasks has now become the industry standard for Natural Language Processing (NLP) models. Pretrained language models, such as BERT (Devlin et al. (2018)), XLNet (Yang et al. (2019)), RoBERTa (Liu et al. (2019)) and ELMo (Peters et al. (2018)) prove how powerful pretrained language models can be. Specifically, BERT is a language model consisting of multiple transformer encoder layers. Transformers (Vaswani et al. (2017)) can capture long-term dependencies between input tokens by using a self-attention mechanism. Self-attention calculates an attention function using three components, query, key, and value, each denoted as matrices Q, K, and V. The attention function is defined as follows:

$$A = \frac{QK^T}{\sqrt{d_k}} \tag{1}$$

$$Attention(Q, K, V) = softmax(A)V \tag{2}$$

It is known that through pretraining using Masked Language Modeling (MLM) and Next Sentence Prediction (NSP), the attention matrices in BERT can capture substantial linguistic knowledge. BERT has achieved the state-of-the-art performance on a wide range of NLP tasks, such as the GLUE benchmark (Wang et al. (2018)) and SQuAD (Rajpurkar et al. (2016)).

However, these modern pretrained models are very large in size and contain millions of parameters, making them nearly impossible to apply on edge devices with limited amount of resources.

**Model Compression.**   As deep learning algorithms started getting adopted, implemented, and researched in diverse fields, high computation costs and memory shortage have started to become challenging factors. Especially in NLP, pretrained language models typically require a large set of parameters. This results in extensive cost of computation and memory. As such, Model Compression has now become an important task for deep learning. There have already been many attempts to tackle this problem, including quantization (Gong et al. (2014)) and weight pruning (Han et al. (2015)). Two promising approaches are KD (Hinton et al. (2015)) and Parameter Sharing, which we focus on in this paper.

**Knowledge Distillation (KD).**   As briefly covered in Section 1, KD transfers knowledge from a well-trained and large teacher model to a smaller student model. KD uses the teacher models predictions on top of the true labels to train the student model. It is proven through many experiments

that the student model learns to imitate the soft distribution of the teacher model's predictions and ultimately performs better than learning solely from the original data. There have already been many attempts to compress BERT using KD. Patient Knowledge Distillation (Sun et al. (2019)) extracts knowledge not only from the final prediction of the teacher, but also from the intermediate layers. TinyBERT (Jiao et al. (2019)) uses a two-stage learning framework and applies KD in both pretraining and task-specific finetuning. DistilBERT (Sanh et al. (2019)) uses half of the layers of BERT-base model and applies KD during pretraining and finetuning of BERT. Insufficient capacity and the absence of a clear initialization guide are some of the existing KD method's area of improvement.

**Parameter Sharing.** Sharing parameters across different layers is a widely used idea for model compression. There have been several attempts to apply parameter sharing in transformer architecture and BERT model. However, the existing parameter sharing methods exhibit a large tradeoff between model performance and model size. They reduce the model's size significantly but also suffers from a great loss in performance as a result.

## 3 PROPOSED METHODS

In the following, we provide an overview of the main challenges faced in KD and our methods to address them in Section 3.1. We then discuss the precise procedures of SPS and PTP in Sections 3.2 and 3.3. Lastly, we explain our final method, PeaBERT and the training details in Section 3.4.

### 3.1 OVERVIEW

BERT-base model contains over 109 million parameters. Its extensive size makes model deployment often infeasible and computationally expensive in many cases, such as on mobile devices. As a result, industry practitioners commonly use a smaller version of BERT and apply KD. However, the existing KD methods face the following challenges:

1. **Insufficient model capacity of the student model.** Since the student model contains fewer number of parameters than the teacher model, its model capacity is also lower. The smaller and simpler the student model gets, the gap between the student and the teacher grows, making it increasingly difficult for the student to replicate the teacher model's accuracy. The limited capacity hinders the student model's performance. How can we enlarge the student model's capacity while maintaining the same number of parameters?
2. **Absence of proper initial guide for the student model.** There is no widely accepted and vetted guide to selecting the student's initial state of the KD process. In most cases, a truncated version of pretrained BERT-base model is used. In reality, this hinders the student from reproducing the teacher's results. Can we find a better method for the student's KD initialization?

We propose the following main ideas to address these challenges:

1. **Shuffled Parameter Sharing (SPS): increasing 'effective' model capacity of the student.** SPS consists of two steps. Step 1 increases the student's effective model capacity by stacking parameter-shared layers. Step 2 further improves the model capacity by shuffling the shared parameters. Shuffling enables more efficient use of the parameters by learning diverse set of information. As a result, the SPS-applied student model achieves much higher accuracy while still using the same number of parameters (details in Section 3.2).
2. **Pretraining with Teacher's Predictions (PTP): a novel pretraining task utilizing teacher's predictions for student initialization.** To address the limitation of the initial guide, we propose PTP, a novel pretraining method for the student by utilizing teacher model's predictions. Through PTP, the student model pre-learns knowledge latent in the teacher's softmax output which is hard to learn during the conventional KD process. This helps the student better acquire and utilize the teacher's knowledge during the KD process (details in Section 3.3).

The following subsections describe the procedures of SPS, PTP, and PeaBERT in detail.

## 3.2 SHUFFLED PARAMETER SHARING (SPS)

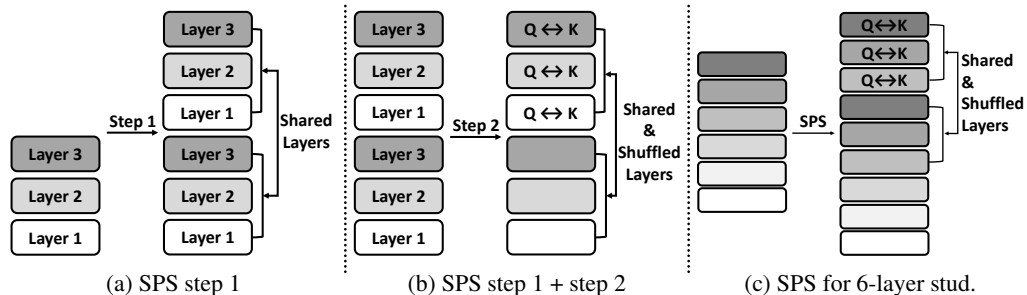

(a) SPS step 1              (b) SPS step 1 + step 2              (c) SPS for 6-layer stud.

Figure 1: Graphical representation of SPS: (a) first step of SPS (b) second step of SPS, and (c) modified SPS for a 6-layer student.

SPS improves student's model capacity while using the same number of parameters, addressing the capacity limitations of a typical KD. SPS is composed of the following two steps.

**Step1. Paired Parameter Sharing.** We start with doubling the number of layers in the student model. We then share the parameters between the bottom half and the upper half of the model, as graphically represented in Figure 1a.

**Step2. Shuffling.** We shuffle the Query and Key parameters between the shared pairs. That is, for the shared upper half of layers we use the original Query parameters as Key parameters, and the original Key parameters as Query parameters.

We call this architecture SPS, which is depicted in Figure 1b. For the 6-layer student case we slightly modify the architecture as shown in Figure 1c. We apply SPS on the top 3 layers only.

**Motivation behind step 1: Increased model capacity due to a more complex structure.** By applying step 1, the student model now has more layers while maintaining the same number of parameters used. This leads to a higher model complexity, which we expect will increase model capacity and performance.

**Motivation behind step 2: Increased parameter efficiency and additional regularization effect.** The intuition behind shuffling becomes clear when we compare the model with and without step 2. Let us consider the case depicted in Figure1b. In step 1, the first layer's Query parameter is used as a Query parameter in both the first layer and in the shared fourth layer. Since this parameter is used only as a Query parameter throughout the model, it will only learn the information relevant to Query. In step 2, on the other hand, this parameter's function changes due to shuffling. The first layer's Query parameter is used as a Key parameter in the shared fourth layer. Thus it has the opportunity to learn the important features of both the Query and the Key, gaining a more diverse and wider breadth of knowledge. The shuffling mechanism allows the parameters to learn diverse information, thereby maximizing learning efficiency. We can also view this as a type of regularization or multi-task learning. Since these parameters need to learn the features of both the Query and the Key, this prevents overfitting to either one. Therefore, we expect the shuffling mechanism to improve the generalization ability of the model.

## 3.3 PRETRAINING WITH TEACHER'S PREDICTIONS (PTP)

There can be several candidates for KD-specialized initialization. We propose a pretraining approach called PTP, and experimentally show that it improves KD accuracy.

Most of the previous studies on KD do not elaborate on the initialization of the student model. There are some studies that use a pretrained student model as an initial state, but those pretraining tasks are irrelevant to either the teacher model or the downstream task. To the best of our knowledge, our study is the first case that pretrains the student model with a task relevant to the teacher model and its downstream task. PTP consists of the following two steps.

**Step 1. Creating artificial data based on the teacher's predictions (PTP labels).**

We first input the training data in the teacher model and collect the teacher model's predictions. We then define "confidence" as the following. We apply softmax function to the teacher model's predictions, and the maximum value of the predictions is defined as the confidence. Next, with a specific threshold "t" (a hyperparameter between 0.5 and 1.0), we assign a new label to the training data according to the rules listed in Table 1. We call these new artificial labels PTP labels.

Table 1: Assigning new PTP labels to the training data.

| Teacher's prediction correct | confidence $> t$ | PTP label |
| :---: | :---: | :---: |
| True | True | confidently correct |
| True | False | unconfidently correct |
| False | True | confidently wrong |
| False | False | unconfidently wrong |

**Step 2. Pretrain the student model to predict the PTP labels.** Using the artificial PTP labels (data $x$, PTP label) we created, we now pretrain the student model to predict the PTP label when $x$ is provided as an input. In other words, the student model is trained to predict the PTP labels given the downstream training dataset. We train the student model until convergence.

Once these two steps are complete, we use this PTP-pretrained student model as the initial state for the KD process.

**Motivation behind PTP: A relatively easy way of learning the teacher's generalized knowledge.** The core idea is to make PTP labels by explicitly expressing important information from the teachers' softmax outputs, such as whether the teacher model predicted correctly or how confident the teacher model is. Pretraining using these labels would help the student acquire the teacher's generalized knowledge latent in the teacher's softmax output. This pretraining makes the student model better prepared for the actual KD process. For example, if a teacher makes an incorrect prediction for a data instance x, then we know that data instance x is generally a difficult one to predict. Since this knowledge is only obtainable by directly comparing the true label with the teachers output, it would be difficult for the student to acquire this in the conventional KD process. Representing this type of information through PTP labels and training the student to predict them could help the student acquire deeply latent knowledge included in the teacher's output much easily. Intuitively, we expect that a student model that has undergone such a pretraining session is better prepared for the actual KD process and will likely achieve better results.

## 3.4 PeaBERT: SPS and PTP combined

### 3.4.1 Overall Framework of PeaBERT

PeaBERT applies SPS and PTP together on BERT for maximum impact on performance. Given a student model, PeaBERT first transforms it into an SPS model and applies PTP. Once PTP is completed, we use this model as the initial state of the student model for the KD process. The overall framework of PeaBERT is depicted in Figure 2.

### 3.4.2 Learning Details of PeaBERT

For the starting point of the KD process, a well-finetuned teacher model should be used. We use the 12 layer BERT-base model as the teacher. The learned parameters are denoted as:

$$\hat{\theta}^t = \arg\min_{\theta^t} \sum_{i \in N} \mathcal{L}_{CE}(y_i, \sigma(z_t(x_i; \theta^t)))$$ (3)

The $\theta^t$ denotes parameters of the teacher, $\sigma$ denotes the softmax function, $x_i$ denotes the training data, $z_t$ denotes the teacher model's output predictions, $y_i$ denotes the true labels, and $\mathcal{L}_{CE}$ denotes cross-entropy loss.

We then pretrain the student model with PTP labels using the following loss:

$$\hat{\theta}^s_{\mathcal{PTP}} = \arg\min_{\theta^s} \sum_{i \in N} \mathcal{L}_{CE}(y_i^{\mathcal{PTP}}, \sigma(z_s(x_i; \theta^s)))$$ (4)

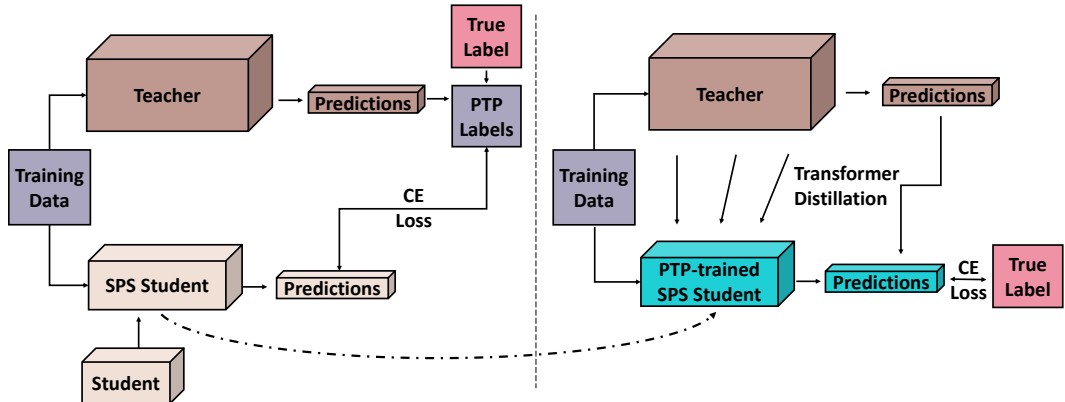

Figure 2: Overall framework of PeaBERT. The left half represents applying SPS and PTP to a student model. The right half represents the learning baseline.

where $y_i^{\mathcal{PTP}}$ denotes the PTP labels and the subscript $s$ denotes the student model. When PTP is completed, we use the $\hat{\theta}_{\mathcal{PTP}}^s$ as the initial state of the KD process. Our KD process uses three loss terms in total: a cross-entropy loss between student model's final predictions and ground-truth labels, a Kullback-Leiber divergence loss between the final predictions of student and teacher models, and a mean-squared error loss between the intermediate layers of student and teacher models.

The loss function is as follows:

$$\mathcal{L} = (1-\alpha)\mathcal{L}_{CE}(y_i, \sigma(z_s(x_i; \theta^s))) + \alpha\mathcal{L}_{\mathcal{KL}}(\sigma(z_s(x_i; \theta^s), \sigma_T(z_t(x_i; \hat{\theta}^t)))) + \beta \sum_{(k_1, k_2) \in \mathcal{K}} ||z_s^{k1} - z_t^{k2}||_2^2$$

(5)

where $\mathcal{L}_{\mathcal{KL}}$ denotes the Kullback-Leibler divergence loss, $\mathcal{K}$ denotes the index pairs of layers we use for intermediate layer-wise distillation, and $z^k$ denotes the output logits of the k-th layer. $\alpha$ and $\beta$ are hyperparameters.

Note that during the KD process, we use a softmax-temperature T, which controls the softness of teacher model's output predictions introduced in (Hinton et al. (2015)).

$$\sigma_T(z_i) = \frac{e^{z_i/T}}{\sum_j e^{z_j/T}}$$

(6)

## 4 EXPERIMENTS

We discuss experimental results to assess the effectiveness of our proposed method. Our goal is to answer the following questions.

- **Q1. Overall performance.** How does PeaBERT perform compared to the currently existing KD methods? (Section 4.2)
- **Q2. Effectiveness of SPS.** To what extent does SPS improve the effective capacity of the student model without increasing the number of parameters? (Section 4.3)
- **Q3. Effectiveness of PTP.** Is PTP a good initialization method? Compared to the conventionally-used truncated version of the BERT-base model, what is the impact of PTP on the model's performance? (Section 4.4)

### 4.1 EXPERIMENTAL SETTINGS

**Datasets.** We use four of the most widely used datasets in the General Language Understanding Evaluation (GLUE) benchmark (Wang et al. (2018)): SST-2[1], QNLI[2], RTE[3], and MRPC[4]. For sentiment classification, we use the Stanford Sentiment Treebank (SST-2) (Socher et al. (2013)). For

---

[1]https://nlp.stanford.edu/sentiment/index.html

[2]https://rajpurkar.github.io/SQuAD-explorer/

[3]https://aclweb.org/aclwiki/RecognizingTextualEntailment

[4]https://www.microsoft.com/en-us/download/details.aspx?id=52398

Table 2: Overall results of PeaBERT compared to the state-of-the-art KD baseline, PatientKD. The results are evaluated on the test set of GLUE official benchmark. The subscript numbers denote the number of independent layers of the student.

| Method | RTE (Acc) | MRPC (F1) | SST-2 (Acc) | QNLI (Acc) | Avg |
|---|---|---|---|---|---|
| BERT-base | 66.4 | 88.9 | 93.5 | 90.5 | 84.8 |
| BERT$_1$-PatientKD | 52.8 | 80.6 | 83.6 | 64.0 | 70.3 |
| PeaBERT$_1$ | 53.0 | 81.0 | 86.9 | 78.8 | 75.0 |
| BERT$_2$-PatientKD | 53.5 | 80.4 | 87.0 | 80.1 | 75.2 |
| PeaBERT$_2$ | 64.1 | 82.7 | 88.2 | 86.0 | 80.3 |
| BERT$_3$-PatientKD | 58.4 | 81.9 | 88.4 | 85.0 | 78.4 |
| PeaBERT$_3$ | 64.5 | 85.0 | 90.4 | 87.0 | 81.7 |

Table 3: PeaBERT in comparison to other state-of-the-art competitors in dev set. The cited results of the competitors are from the official papers of each method. For accurate comparison, model dimensions are fixed to six layers across all models compared.

| Method | # of parameters | RTE (Acc) | MRPC (F1) | SST-2 (Acc) | QNLI (Acc) | Avg |
|---|---|---|---|---|---|---|
| DistilBERT | 42.6M | 59.9 | 87.5 | 91.3 | 89.2 | 82.0 |
| TinyBERT | 42.6M | 70.4 | 90.6 | 93.0 | 91.1 | 86.3 |
| PeaBERT | 42.6M | 73.6 | 92.9 | 93.5 | 90.3 | 87.6 |

natural language inference, we use QNLI(Rajpurkar et al. (2016)) and RTE. For paraphrase similarity matching, we use Microsoft Research Paraphrase Corpus (MRPC) (Dolan & Brockett (2005)). Specifically, SST-2 is a movie review dataset with binary annotations where the binary label indicates positive and negative reviews. QNLI is a task for predicting whether a pair of a question and an answer is an entailment or not. RTE is based on a series of textual entailment challenges and MRPC contains pairs of sentences and corresponding labels, where the labels indicate the semantic equivalence relationship between the sentences in each pair.

**Competitors.** We use Patient Knowledge Distillation (PatientKD, Sun et al. (2019)) as our baseline learning method to compare and quantify the effectiveness of PeaBERT. PatientKD is one of the most widely used baselines and is a variant of the original Knowledge Distillation method (Hinton et al. (2006)). We conduct experiments on BERT model (Devlin et al. (2018)) and compare the results of PeaBERT to the original BERT. In addition, we compare the results with other state-of-the-art BERT-distillation models, including DistilBERT(Sanh et al. (2019)) and TinyBERT(Jiao et al. (2019)).

**Training Details.** We use the 12-layer original BERT model (Devlin et al. (2018)) as a teacher model and further finetune the teacher for each task independently. The student models are created using the same architecture as the original BERT, but the number of layers is reduced to either 1, 2, 3, or 6 depending on experiments. That is, we initialize the student model using the first $n$-layers of parameters from the pretrained original BERT obtained from Google's official BERT repo[5]. We use the standard baseline PatientKD and the following hyperparameter settings: training batch size from $\{32, 64\}$, learning rate from $\{1, 2, 3, 5\} \cdot 10^{-5}$, number of epochs from $\{4, 6, 10\}$, $\alpha$ between $\{0.1 \text{ and } 0.7\}$, and $\beta$ between $\{0 \text{ and } 500\}$.

## 4.2 Overall Performance

We summarize the performance of PeaBERT against the standard baseline PatientKD in Table 2. We also compare the results of PeaBERT to the competitors DistilBERT and TinyBERT in Table 3. We observe the following from the results.

First, we see from Table 2 that PeaBERT consistently yields higher performance in downstream tasks across all three model sizes. Notably, PeaBERT shows improved accuracy by an average of 4.7% for the 1-layer student, 5.1% for the 2-layer student, and 3.3% for the 3-layer student (Note that we use F1 score for MRPC when calculating the average accuracy throughout the experiments).

---

[5]https://github.com/google-research/bert

A maximum improvement of 14.8% is achieved in PeaBERT$_1$-QNLI. These results validate the effectiveness of PeaBERT across varying downstream tasks and student model sizes.

Second, using the same number of parameters, PeaBERT outperforms the state-of-the-art KD baselines DistilBERT and TinyBERT, by 5.6% and 1.3% on average. We use a 6-layer student model for this comparison. An advantage of PeaBERT is that it achieves remarkable accuracy improvement just by using the downstream dataset without touching the original pretraining tasks. Unlike its competitors DistilBERT and TinyBERT, PeaBERT does not touch the original pretraining tasks, Masked Language Modeling (MLM) and Next Sentence Prediction (NSP). This reduces training time significantly. For example, DistilBERT took approximately 90 hours with eight 16GB V100 GPUs, while PeaBERT took a minimum of one minute (PeaBERT$_1$ with RTE) to a maximum of one hour (PeaBERT$_3$ with QNLI) using just two NVIDIA T4 GPUs.

Finally, another advantage of PeaBERT is that it can be applied to other transformer-based models with minimal modifications. The SPS method can be directly applied to any transformer-based models, and the PTP method can be applied to any classification task.

### 4.3 Effectiveness of SPS

Table 4: An ablation study to validate each step of SPS. The results are derived using GLUE dev set.

| Method | # of parameters | RTE (Acc) | MRPC (F1) | SST-2 (Acc) | QNLI (Acc) | Avg |
|---|---|---|---|---|---|---|
| BERT$_3$ | 21.3M | 61.4 | 84.3 | 89.4 | 84.8 | 80.0 |
| SPS-1 | 21.3M | 63.5 | 85.8 | 89.6 | 85.5 | 81.1 |
| SPS-2 | 21.3M | 68.6 | 86.8 | 90.2 | 86.5 | 83.0 |

We perform ablation studies to verify the effectiveness of SPS. We compare three models BERT$_3$, SPS-1, and SPS-2. BERT$_3$ is the original BERT model with 3 layers, which applies none of the SPS steps. SPS-1 applies only the first SPS step (paired parameter sharing) to BERT$_3$. SPS-2 applies both the first step and the second step (shuffling) to BERT$_3$.

The results are summarized in Table 4. Compared to the original BERT$_3$, SPS-1 shows improved accuracy in all the downstream datasets with an average of 1.1%, verifying our first motivation. Comparing SPS-1 with SPS-2, we note that SPS-2 consistently shows even better performance with an average of 1.9%, which validates our second motivation. Based on these results, we conclude that both steps of the SPS process work as intended to increase the student model's capacity without increasing the number of parameters used.

### 4.4 Effectiveness of PTP

Table 5: An ablation study to verify PTP. The results are derived using GLUE dev set.

| Model | RTE (Acc) | MRPC (F1) | SST-2 (Acc) | QNLI (Acc) | Avg |
|---|---|---|---|---|---|
| BERT$_3$+SPS | 68.6 | 86.8 | 90.2 | 86.5 | 83.0 |
| BERT$_3$+SPS+PTP-1 | 69.0 | 88.0 | 90.4 | 86.6 | 83.5 |
| BERT$_3$+SPS+PTP-2 | 69.3 | 88.3 | 90.9 | 86.8 | 83.8 |
| BERT$_3$+SPS+PTP | 70.8 | 88.7 | 91.2 | 87.1 | 84.5 |

We perform an ablation study to validate the effectiveness of using PTP as an initial guide for the student model. We use BERT$_3$+SPS, which is SPS applied on BERT$_3$, as our base model. We compare the results of PTP to its variants PTP-1 and PTP-2. Note that PTP uses four labels constructed from two types of information latent in the teacher model's softmax prediction: (1) whether the teacher predicted correctly and (2) how confident the teacher is. PTP-1 is a variant of PTP that uses only two labels that state whether the teacher predicts correctly or not. PTP-2 is another PTP variant that uses two labels to state whether the teacher's predictions is confident or not. PTP-1 and PTP-2 contain different type of teacher's information in their labels, respectively. From the results summarized in Table 5, we see that PTP-1 and PTP-2 increase the student's accuracy by 0.5% and 0.8% on average, respectively. Combining PTP-1 and PTP-2, which is essentially our proposed PTP method, further improves model accuracy by an average of 1.5%. This study supports the efficacy of our PTP method.

These results prove two things: (1) The student that goes through only the conventional KD process does not fully utilize the knowledge included in the teacher models softmax outputs, and (2) PTP does help the student better utilize the knowledge included in teachers softmax outputs. This firmly supports the efficacy of our PTP method and also validates our main claim that initializing a student model with KD-specialized method prior to applying KD can improve accuracy. As existing KD methods do not place much emphasis on the initialization process, this finding highlights a potentially major, undiscovered path to improving model accuracy. Further and deeper researches related to KD-specialized initialization could be promising.

## 5 CONCLUSION

In this paper, we propose Pea-KD, a new KD method for transformer-based distillation, and show its efficacy. Our goal is to address and reduce the limitations of the currently available KD methods: insufficient model capacity and absence of proper initial guide for the student. We first introduce SPS, a new parameter sharing approach that uses a shuffling mechanism, which enhances the capacity of the student model while using the same number of parameters. We then introduce PTP, a KD-specific initialization method for the student model. Our proposed PeaBERT comes from applying these two methods SPS and PTP on BERT. Through extensive experiments conducted using multiple datasets and varying model sizes, we show that our method improves KD accuracy by an average of 4.4% on the GLUE test set. We also show that PeaBERT works well across different datasets, and outperforms the original BERT as well as other state-of-the-art baselines on BERT distillation by an average of 3.5%.

As a future work, we plan to delve deeper into the concept of KD-specialized initialization of the student model. Also, since PTP and SPS are independent processes on their own, we plan to combine PTP and SPS with other model compression techniques, such as weight pruning and quantization.

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
