# OpenReview forum: "Pea-KD: Parameter-efficient and accurate Knowledge Distillation"
_ICLR.cc/2021/Conference — Reject_

### Official Review · AnonReviewer4 · 2020-10-14
**Interesting ideas, but each idea seems only weakly motivated to me; inference times are not mentioned or measured.**

**Rating:** 6
**Confidence:** 4

**Review:**

The paper claims to present an improved form of knowledge distillation which tackles the following perceived weaknesses of existing kd systems:
- student's expressive power is less than original teacher model, and
- unclear how to initialize the student model weights in a principled way

The paper proposes two methods to improve kd:
- using stacked layers, with weight sharing between the weights, and keys and queries swapped in the upper layers
   - this is proposed because it means the effective student geometry matches that of the teacher, or at least is significantly larger than without the shared layers, whilst the number of unique parameters remains small
- getting the student to predict the confidence of the teacher on each example, rather than just the softmax output of the teacher
   - the confidence is classified as either 'strong' or 'weak'
   - the student must also predict whether the teacher gets the example correct or not (which we know, since this is for training set examples, for which we have ground truth)

Strong points of the paper:
- SPS sounds novel, but I felt it was only very weakly motivated
    - in addition it sounds to me like the inference time will be similar to the original model? I feel that reducing the number of parameters in the student is not the only goal: the student model should be fast to execute.
    - the paper does admittedly not claim to provide fast inference times as a goal; nevertheless I have a hard time imagining a BERT model running on an edge device with a limited amount of resources in a reasonable time, so I think that inference time is a critical metric which I feel this paper does not consider, or measure
- PTP sounds to me only weakly novel, since typically the student will be trained to predict the full softmax output of the teacher, whereas in PTP, the student must predict whether the highest value of the softmax is high or low. I feel that little insight or motivation is given as to why this auxiliary task was used, or is better than predicting the full softmax distribution. Nor was a rigorous comparison carried out to compare using just this pre-training approach with just the softmax-matching approach, where both using appropriately tuned hyper-parameters, I feel. There was an experiment in 4.4 where the PTP task is dropped, but I got the impression that this was after tuning the hyper-parameters to rely on PTP task? The hyper-parameters would I feel need to be retuned in the absence of the PTP task I think?

I mostly like the paper. I found it interesting. However, I have two main concerns:
- the approaches used in the paper feel to me only weakly motivated. Little insight is given into why the approaches were chosen, and why they should work. For example, flipping the keys and queries was not well justified I felt. Nor was the PTP task relative to traditional softmax matching.
- the paper avoids discussing inference times, and it's unclear to me whether a distillation approach that results in few parameters but long inference times would be useful in practice?
    - of course, the underlying techniques can still be interesting from a theoretical point of view

Whilst I was reading through the paper, I wrote down some more detailed reactions:

Introduction:

"stacking the layers that share parameters" => "stacking layers that share parameters" (the former form of language is using 'that share parameters' as a filter to choose from existing layers; the latter form implies that we create new layers, which we then stack, and which we configure to have shared parameters)

3.1 Overview

So, it looks like SPS means the number of unique parameters is kept low whilst the effective model complexity is unchanged compared to original BERT. How does this affect inference speed? It sounds to me like the inference speed under SPS will be unchanged from a BERT model of equivalent model complexity? Often, a use-case of distillation is to reduce inference time, and requirements in terms of power, eg for mobile devices, or production inference. Assuming that the inference time is unchanged from the original model of equal complexity, what is the target use-case of SPS/Pea-KD?

3.2 SPS

No insight or intuition is given as to why swapping query and key is likely to be a useful thing to do. I think it would be good to provide such insight and intuition. Like, you say that it increases the expressive power, but you don't justify why you feel this is true.

For the six-layer student, I couldnt quite undersatnd why it looks like the 6-layer student has 9 layers? Please consider clarifying this point:
- why do you call it '6 layer' if it has 9 effective layers? Perhaps consider renaming it to 9-layer student?
- why do you make the 6 layer model become 9 layers, and not 12 layers? ie, following the initial recipe detailed in section 3.2, of doubling all the layers.

3.3 PTP

I feel that the column for 'PTP label' in table 1 should be on the right hand side:
- the inputs should be I feel on the left, ie 'teachers prediction correct', and 'confidence > t',
- and then we can read off the 'output' of the table as the right-hand column
- (currently I find the table hard to read)

The pretraining task itself seems interesting to me.

How do you get the student to predict this result? Do you put a linear layer on the output from the student? Why do you use a classification task (4 classes) instead of using a regression task to predict the confidence? did you try a regression task to predict the confidence? Why use a classification into 4 classes, rather than two classifiers into two classes each?

4. Experiments:

For table 2, I wasn't really sure how these numbers compare to two key baselines:
- a simple bilstm with attention, without any pretraining, and
- a BERT-base model

So, I dug these out, and here are the peabert numbers from table 2, put into the context of these two baselines (which I feel are lower and upperbound really for what we'd expect to see from PeaBERT):

     BiLSTM+Attn
     (single-task training)   BERT-base   PeaBERT1  PeaBERT2  PeaBERT3
    RTE: 51.9                     66.4        53.0     64.1       64.5
    MRPC 68.5                     88.9        81.0     82.7        85.0
    SST2 85.9                     93.5        86.9     88.2       90.4
    QNLI 77.2                     90.5        78.8      86.0       87.0

I got the BiLSTM+Attn numbers from GLUE paper, and
BERT numbers from https://arxiv.org/pdf/1910.03176.pdf

I kind of feel that I might not be the only person who might want to see the PeaBERT numbers in the context of such lower and upperbound baselines? Maybe consider adding these baselinse into table 2?

Basically, my take away from this is that PeaBERT1 is barely better than the simple BiLSTM+Attn baseline, but PeaBERT3 approaches the score of a full BERT-base model?

I kind of think the sentence "For example, DistilBERT took approximately 90 hours with eight 16GB V100 GPUs while PeaBERT took a minimum of one minute (PeaBERT1 with RTE) to a maximum of one hour (PeaBERT3 with QNLI) using just two NVIDIA T4 GPUs" should be highlighted in a table somewhere somehow perhaps, rather than buried in text? Not sure if that's a good idea, just occurred to me though.

Question: why only show results on a subset of the GLUE tasks? eg patient-kd paper shows results on additionally:
- QQP
- MNLI-m
- MNLI-mm
(whilst also showing results for: SST-2, MRPC, QNLI and RTE, as here)

It is unclear from this whether you ran against all, and only showed the four tasks that show a benefit, or whether you simply didn't have time to run on all 7 tasks. Preference to show results for all 7 tasks that Patient-KD paper reports results for.

For ablation studies, I feel it would be interesting to see an ablation study that removes each of various losses in equation 3 in turn.

Importantly, none of the experiments mention inference time, which I feel is a key metric to report for distillation?

---

> ### Author Response · Authors · 2020-11-24
> **Response to Reviewer # 4 - general thanks and comments (1/3)**
>
>
> [Comment 1]  "the approaches used in the paper feel to me only weakly motivated. Little insight is given into why the approaches were chosen, and why they should work. For example, flipping the keys and queries was not well justified I felt. Nor was the PTP task relative to traditional softmax matching."
>
> => SPS Step 2 (Shuffling)
>
> We believe the main factor contributing to the power of SPS step 2 is as follows:
>
> Maximizing features learned by parameters by taking on different roles within the model.
>
> Taking the BERT$_{3}$ case as an example, let us look at the difference between SPS-1 and SPS-2 and how that contributes to the learning process of the parameters. As a recap, while SPS-1 applies only step1 to BERT, SPS-2 applies both step 1 and step 2, shuffling the parameters around.
> Consider the first layer's Query parameter. Under SPS-1, this parameter is used as a Query parameter in both the first layer and in the shared fourth layer. Since this parameter is used only as Query, it will learn only the information relevant to Query.
> Under SPS-2, however, the first layer’s Query parameter’s function changes due to shuffling. The first layer’s Query parameter is used as a Key parameter in the shared fourth layer. This one parameter has had the opportunity to learn the important features of both the Query and the Key functions, gaining a more diverse and wider breadth of knowledge. Based on the average accuracy increase of 1.9 percent driven by shuffling (Table 4 of the paper), we believe that the shuffled parameters were able to learn a more diverse set of information, and this improvement in parameter efficiency contributed to enriching model capacity. We also see this in a similar but slightly different point of view. Since the parameters get to learn diverse features and get to function in diverse positions we believe this could act as an additional regularization. Therefore it helps the model prevent overfitting and leads to improvement in performance.
>
> => PTP
>
> The core idea is to make PTP labels by explicitly expressing important information from the teachers’ softmax outputs, such as whether the teacher model predicted correctly or how confident the teacher model is. Pretraining using these labels would help the student acquire the teacher’s generalized knowledge latent in the teacher’s softmax output more easily. This pretraining makes the student model better prepared for the actual KD process. For example, if a teacher makes an incorrect prediction for a data instance x, then we also know that x is generally a difficult one to predict. Since this knowledge is obtainable only by directly comparing the true label with the teachers output, it would be difficult for the student to acquire this in the conventional KD process. Representing this type of information through PTP labels and training the student to predict them could help the student acquire deeply latent knowledge included in the teacher’s output much easily. Intuitively, we expect that a student model that has undergone such a pretraining session is better prepared for the actual KD process and will likely achieve better results.
>
> [Comment 2 & 3]  "the paper avoids discussing inference times, and it's unclear to me whether a distillation approach that results in few parameters but long inference times would be useful in practice?"
>
> •	 of course, the underlying techniques can still be interesting from a theoretical point of view
>
> &
>
> So, it looks like SPS means the number of unique parameters is kept low whilst the effective model complexity is unchanged compared to original BERT. How does this affect inference speed? It sounds to me like the inference speed under SPS will be unchanged from a BERT model of equivalent model complexity? Often, a use-case of distillation is to reduce inference time, and requirements in terms of power, eg for mobile devices, or production inference. Assuming that the inference time is unchanged from the original model of equal complexity, what is the target use-case of SPS/Pea-KD?
>
> => Our primary focus is on improving performance without needing additional memory resources. Of the many important factors that we consider in model compression, such as memory storage, performance, and inference time, some factors are prioritized over others depending on the circumstances. In our paper, we prioritized maximum performance improvement while keeping memory constant, over speed. As you have pointed out, the SPS method has a drawback of additional inference time. However, despite the longer inference time, our model does perform significantly better. We believe that the additional time could be acceptable in certain cases where the performance is much more important than inference time, with limited memory storage. We do acknowledge that the increased inference time is an important limitation of our SPS method and will certainly aim to reduce this in our future work.
>
> (Continued)

---

> > ### Author Response · Authors · 2020-11-24
> > **Response to Reviewer # 4 - general thanks and comments (2/3)**
> >
> >
> > (continued)
> >
> > As you mentioned, despite the drawback of greater inference time, the underlying techniques of SPS provide a new approach to thinking about Parameter Sharing. To the best of our knowledge, this is the first approach in Parameter Sharing that uses shared internal parameters in different positions in order to train parameters more efficiently. We believe SPS is meaningful, because it has shown considerable and consistent performance improvement in a relatively intuitive manner. Due to its simplicity, we expect SPS to be widely applicable to many models that apply parameter sharing.
> >
> > [Comment 4]
> > Introduction:
> > "stacking the layers that share parameters" => "stacking layers that share parameters" (the former form of language is using 'that share parameters' as a filter to choose from existing layers; the latter form implies that we create new layers, which we then stack, and which we configure to have shared parameters)
> >
> > => Thank you for the correction. We updated the statement accordingly.
> >
> > [Comment 5]
> > " 3.3 PTP
> > I feel that the column for 'PTP label' in table 1 should be on the right hand side:
> > •	the inputs should be I feel on the left, ie 'teachers prediction correct', and 'confidence > t',
> > •	and then we can read off the 'output' of the table as the right-hand column
> > •	(currently I find the table hard to read) "
> >
> > =>  Thank you. We made modifications based on your comments.
> >
> > [Comment 6] "The pretraining task itself seems interesting to me.
> > How do you get the student to predict this result? Do you put a linear layer on the output from the student?"
> >
> > => We used a linear layer on top of BERT to classify the PTP labels.
> >
> > [Comment 7] "Why do you use a classification task (4 classes) instead of using a regression task to predict the confidence? did you try a regression task to predict the confidence? Why use a classification into 4 classes, rather than two classifiers into two classes each?"
> >
> > => The starting point of creating the PTP labels focused on the teacher model’s correctness. We initially started with just one set of binary labels denoting teacher's correctness. It was not until later on in our research that we came up with the idea of adding in confidence-based labels. This is how we decided on the four classes.
> > Using a regression task also seems like a great idea. However, we would be predicting only the level of confidence not the correctness. Because we started from correctness labels, we had not thought of using regression yet. We did not have enough time to incorporate this feedback into our paper but will certainly try this for future research. The same for using two classifiers also. We will try this in the future work.
> >
> > [Comment 8] "I kind of feel that I might not be the only person who might want to see the PeaBERT numbers in the context of such lower and upperbound baselines? Maybe consider adding these baselinse into table 2?"
> >
> > => We think it is a good idea. Since we are working on BERT compression, adding in BERT-base as baseline will make the comparison more robust. We have added this to Table 2 in section 4.2 of our paper. Thank you for the feedback.
> >
> > [Comment 9] "Basically, my take away from this is that PeaBERT1 is barely better than the simple BiLSTM+Attn baseline, but PeaBERT3 approaches the score of a full BERT-base model?"
> >
> > => We are not sure if we agree with this comment. Even though PeaBERT3 performs much better than BERT3 we do not agree with the opinion that PeaBERT3 reaches the score of full BERT-base model. The performance gap between PeaBERT3 and BERT-base is 1.9\% for RTE, 3.9\% for MRPC, 3.1\% for SST2, and 3.5\% for QNLI. From our experience, once BERT model reaches a certain level of accuracy, it is extremely difficult to achieve even one percent of improvement, so we believe the current gap between our PeaBERT3 and BERT-base is acceptable.
> >
> > [Comment 10] "why only show results on a subset of the GLUE tasks?"
> >
> > => There are several reasons why we assume the four datasets are actually quite sufficient for our case. First of all, PeaBERT uses both SPS and PTP to improve its performance. However, to apply PTP it has to be a classification task since the labels are constructed from the information of whether the teacher predicts correctly or not. QQP is not a classification task so we excluded it. Also, for MNLI-m and MNLI-mm they are similar to the tasks QNLI and RTE that they are all Natural Language Inference tasks. Therefore, we chose QNLI and RTE instead of MNLI considering the dataset size. However, we appreciate your suggestion to examine the method with more tasks and will try our best to do additional experiments.

---

> > > ### Author Response · Authors · 2020-11-24
> > > **Response to Reviewer # 4 - general thanks and comments (3/3)**
> > >
> > >
> > > [Comment 11 & 12] "For ablation studies, I feel it would be interesting to see an ablation study that removes each of various losses in equation 3 in turn." & " Nor was a rigorous comparison carried out to compare using just this pre-training approach with just the softmax-matching approach, where both using appropriately tuned hyper-parameters, I feel."
> > >
> > > => In your comments, you had suggested including an ablation study that removes each loss terms in the equation (3) (equation (5) in the revised version of the paper). This would be comparing PatientKD, KD, and finetuning (FT) baselines. We did not do this study by ourselves, because this study is included in the Patient Knowledge Distillation paper. We thought it would not add much value to replicate this study. Rather, we thought it would be insightful to compare (a) 'BERT+PKD' with (b) BERT+PTP+FT'. Since the teacher’s knowledge is not used in the finetuning process of (b), we were able to examine whether PTP is fulfilling our intention of helping the student learn the teacher’s generalized knowledge. The results are attached below, where the scores are measured by an average of 5 runs with random seeds for each dataset.
> > >
> > > From the results summarized in the table below, we can see that applying PTP and then finetuning gives better results than the conventional PatientKD by a margin of 0.5\% in accuracy.
> > >
> > > This supports that PTP is a better way of using the teacher’s softmax output than the conventional KD baselines!
> > >
> > > This result also verifies that PTP works exactly as our motivation: student can learn the generalized teacher’s knowledge more easily.
> > >
> > >
> > > |models|MRPC|RTE|SST-2|QNLI|AVG|
> > > |:----|:----|:---|:----|:-----|:-----|
> > > |BERT$_{3}$+PatientKD|   84. 6| 61.9|88.6|84.9|80.0|
> > > |BERT$_{3}$+PTP+FT| 85.3 | 62.5 | 89.4 | 84.9| 80.5|

---

### Official Review · AnonReviewer2 · 2020-10-26
**The paper is poorly written. The motivation and the proposed method are problematic.**

**Rating:** 5
**Confidence:** 4

**Review:**

This paper proposes a distillation method for BERT. The work is based on two-fold main ideas. First, as the student model is usually smaller in the number of parameters, the model capacity is limited. The authors propose to stack the layers that share parameters to counter this limitation. Second, the authors argue that the initialization of the student model is crucial, so they propose an pre-training strategy for boosting the student's performance.

Pros:
The idea of learning good initializations for the student model is interesting.


Cons:
This paper has several writing problems which need be carefully addressed before its publication.

- As all the experiments are conducted on GLUE and only BERT models are considered in the paper, the proposed method seems to be tailored for BERT. The authors should stress this point clearly and early in the paper (in the title, abstract or introduction). Otherwise, the authors should provide some experimental results on other models or tasks, e.g., the typical image recognition task in computer vision.

- In the section of PROPOSED METHODs (section 3.1), the motivation and the main idea of the paper are introduced again. As no any new information is provided here compared to the abstraction and the introduction sections, it seems very redundant.

- The introduction of SPS in Section 3.2 is quite confusing. What do Key  and Query parameters stand for?  I have no idea what the author is talking about here. Maybe it is because I have little background in NLP and BERT. However, even so the authors are still responsible for making the paper easy to follow for readers like me.


As for the motivation and the proposed method, my comments are as follows.

- The motivation and the proposed method are somewhat problematic. Firstly, the authors argue that small student with few parameters are limited in capacity, so they propose to stack repeating layers to enlarge the model capacity. However, stacking repeating layer will introduce much more computation cost, which violates the goal of distillation.

-  Secondly, the authors propose to pre-train the student model with the teacher predictions to initialize the student. However, it is odd to view this step as pre-training as it actually adopts the teacher predictions to train the student. It is actually doing distillation! The improvement  in performance may simply come from the more training epochs.

======================

post-rebuttal:

I have read all the comments from other reviewers and replies from the authors. The revised version partially addresses my concerns so I raise the score from 4 to 5. However, my concerns about the motivation of the work still exist, so I am still slightly leaning to reject this paper.

---

> ### Author Response · Authors · 2020-11-24
> **Response to Reviewer # 2 - general thanks and comments (1/2)**
>
> We would like to thank Reviewer 2 for the insightful comments.
> Each comment and how we have addressed them in our paper is summarized below.
>
> [Comment 1] “As all the experiments are conducted on GLUE and only BERT models are considered in the paper, the proposed method seems to be tailored for BERT. The authors should stress this point clearly and early in the paper (in the title, abstract or introduction). Otherwise, the authors should provide some experimental results on other models or tasks, e.g., the typical image recognition task in computer vision.”
>
> => Based on your comment, we have modified our title to explicitly mention BERT. The new title is “Pea-KD: Parameter-efficient and accurate Knowledge Distillation on BERT.” We also clarified in our paper that these experiments were only conducted on BERT.
>
> [Comment 2] “In the section of PROPOSED METHODs (section 3.1), the motivation and the main idea of the paper are introduced again. As no any new information is provided here compared to the abstraction and the introduction sections, it seems very redundant.”
>
> => 	We apologize if our paper structure was not concise. Since we are introducing a multi-step procedure, we thought it would help the reader follow the paper better if we provided a quick recap of our main ideas mentioned in the introduction once again in Section 3.1. Since the overview and the introduction are both intended to provide a general summary of the proposed method, we hope you will allow for some overlap. To differentiate the introduction from Section 3.1, we have added more information based on multiple Reviewers’ feedback, including discussions around the motivation and intuition behind how we came up with the process, why we intuitively thought this method will work, and the contributing factors to the increased model representation. We have also added more information about SPS and PTP in Section 3.1 based on other reviewers’ feedback.
>
> [Comment 3] “The introduction of SPS in Section 3.2 is quite confusing. What do Key and Query parameters stand for? I have no idea what the author is talking about here. Maybe it is because I have little background in NLP and BERT. However, even so the authors are still responsible for making the paper easy to follow for readers like me.”
>
> => We apologize for not providing enough background on the jargons used. We updated the first paragraph in Section 2 to include explanations of the transformer layers.
>
> [Comment 4] “The motivation and the proposed method are somewhat problematic. Firstly, the authors argue that small student with few parameters are limited in capacity, so they propose to stack repeating layers to enlarge the model capacity. However, stacking repeating layer will introduce much more computation cost, which violates the goal of distillation.”
>
> => Our primary focus was on improving performance without needing additional memory resources. Of the many important factors that we consider in model compression, such as memory storage, performance, and inference time, some factors are prioritized over others depending on the circumstances. In our paper, we prioritized maximum performance improvement while keeping memory constant, over speed.
> As you have pointed out, the SPS method has a drawback of additional inference time. Despite the longer inference time, our model does perform significantly better. We believe that the additional time could be acceptable in certain cases where the performance is much more important than inference time, with limited memory storage. We do acknowledge that the increased inference time is an important limitation of our SPS method and will certainly aim to reduce this in our future work.

---

> > ### Author Response · Authors · 2020-11-24
> > **Response to Reviewer # 2 - general thanks and comments (2/2)**
> >
> > [Comment 5] “Secondly, the authors propose to pre-train the student model with the teacher predictions to initialize the student. However, it is odd to view this step as pre-training as it actually adopts the teacher predictions to train the student. It is actually doing distillation! The improvement in performance may simply come from the more training epochs.”
> >
> > => Clarification of the factors contributing to performance improvement in PTP.
> >
> > During PTP, we are training the student with artificial labels created from the teacher’s output.
> > We are not directly matching the student’s output to the teacher’s output, which is what is done in conventional Knowledge Distillation.  We understand why PTP might seem like distillation depending on how “distillation” is defined. PTP can loosely be understood as a type of distillation, but it is different from and outperforms the conventional distillation methods available.
> > Even when interpreted as a type of distillation, PTP still meaningfully contributes to performance improvement.
> >
> > To address your comment about the performance improvement possibly coming from simply training over a greater number of epochs, we have conducted an experiment as outlined below:
> > To support our claim, we ran experiments comparing two cases. (a) 'BERT + 8 epoch training' and (b) 'BERT$_{3}$+ 2 epoch PTP + 6 epoch training' where training is done with PatientKD. Both models have the same number of total epochs of 8.  The results are the average of 5 runs with random seeds for each dataset.
> > According to the results reported below, (b) shows better accuracy of 0.7\% on average. This shows that increasing the number of epochs is not the main cause of the improvement in PTP. Therefore, we strongly believe that PTP has its own merits.
> >
> > We have also conducted an additional ablation study regarding PTP in the first paragraph of Section 4.4 of the revised paper, in order to further verify the effectiveness of PTP.
> > We would truly appreciate it if you could review our updated paper once more. Thank you.
> >
> > |models|MRPC|RTE|SST-2|QNLI|AVG|
> > |:----|:----|:---|:----|:-----|:-----|
> > |(a) BERT$_{3}$+training(8 epoch)| 85.1 | 62.5 | 88.4 | 85.3| 80.3|
> > |(b) BERT$_{3}$+PTP(2 epoch)+training(6 epoch)|   85.5| 64.0|88.9|85.6|81.0|

---

### Official Review · AnonReviewer3 · 2020-10-28
**Review of AnonReviewer3**

**Rating:** 5
**Confidence:** 3

**Review:**

This paper proposes a parameter-efficient KD which consists of two main parts: Shuffled Parameter Sharing (SPS) and Pretraining with Teacher’s Predictions (PTP). This work explores some new framework like parameters-sharing and shuffling in KD and obtain promising results.


Strengths:
1. The paper is well-written and easy to follow.
2. The experiment part explores the overall performance, the effects of SPS, and PTP separately, which is clear and makes readers easy to understand the effects of each part (SPS: step1+step2, PTP) in Pea-KD.

Weaknesses:
1. When applying SPS, the number of layers in the student model is double than the normal case, even the parameters are the same as the counterpart, I guess the FLOPs will increase or be doubled than the original one. I think for a small student model, the number of parameters should not be the only metric but also consider the FLOPs. Why this paper doesn’t discuss this? This is my main concern.
2. From Table4, we can find that the shuffling in SPS has great effects. step1+step2 would have much greater improvement than only applying step1, how about only conduct shuffling without parameters sharing (only step2 by shuffling the original parameters without step1)?  On the other hand, what is the potential reason why shuffling can enrich the model capacity?

---

> ### Author Response · Authors · 2020-11-24
> **Response to reviewer #3 - general thanks and comments (1/2)**
>
> We would like to thank Reviewer 3 for the insightful and detailed feedback. The order of responses to your questions is intentionally switched for easier explanation.
>
> 1. Potential reasons for why shuffling can enrich model capacity
>
> We believe the main factor contributing to the power of SPS step 2 is as follows:
>
> Increased model capacity by learning diverse set of information with the same parameters.
>
> Taking the BERT$_{3}$ case as an example, let us look at the difference between SPS-1 and SPS-2 and how that contributes to the learning process of the parameters. As a recap, while SPS-1 applies only step1 to BERT, SPS-2 applies both step 1 and step 2, shuffling the parameters around.
> Consider the first layer's Query parameter. Under SPS-1, this parameter is used as a Query parameter in both the first layer and in the shared fourth layer. Since this parameter is only used as Query, it will only learn the information relevant to Query.
> Under SPS-2, however, the first layer’s Query parameter’s function changes due to shuffling. The first layer’s Query parameter is used as a Key parameter in the shared fourth layer. This one parameter has had the opportunity to learn the important features of both the Query and the Key functions, gaining a more diverse and wider breadth of knowledge. Based on the average accuracy increase of 1.9 percent driven by shuffling (Table 4 of the paper), we believe that the shuffled parameters were able to learn a more diverse set of information, and this improvement in parameter efficiency contributed to enriching model capacity. We also see this in a similar but slightly different point of view. Since the parameters get to learn diverse features and get to function in diverse positions we believe this could act as an additional regularization. Therefore it helps the model prevent overfitting and leads to improvement in performance.
>
> 2. Regarding only applying the SPS step 2 to the student model.
>
> We initially did not try applying SPS step 2 on a standalone basis, because we created step 2 with the intention of applying it to the shuffled parameters derived from step 1. We believe step 2 would be meaningful only when preceded by step 1. Since Query and Key are identical (768, 768) linear layers with different names, the performance of BERT and BERT with only step 2 applied would yield the same results. Without step 1, step 2 would not make an impact.
> However, thanks to reviewer 3, we realized that, since we are using the pre-trained BERT supplied by Huggingface, BERT and `BERT + only step 2' could potentially yield different results in this case.
> Given the pre-trained Query and Key parameters by Huggingface, we switched the order of them and performed finetuning. This experiment of using 'BERT + only step 2' showed that applying step 2 alone actually showed decreased average accuracy of 0.6\%. As we expected, step1 needs to be preceded for step2 to have desirable effects.
>
> |models|MRPC|RTE|SST-2|QNLI|AVG|
> |:----|:----|:---|:----|:-----|:-----|
> |BERT$_{3}$|   84. 7| 62.0|88.3|85.1|80.0|
> |BERT$_{3}$+step2| 84.8 | 61.9 | 87.4 | 83.5| 79.4|
>
> 3. Regarding the increased inference time.
>
> Compared to the original BERT model, our SPS model uses the same amount of memory storage to load and run but achieves much higher accuracy, an average improvement of 4.4\% . However, we are aware and fully acknowledge that our approach has a drawback of additional inference time. This is mostly because we use additional shared layers as the reviewer supposed, which takes about 50\% to 100\% additional time to run for PeaBERT (50\% for PeaBERT6 and 100\% for PeaBERT3). Despite the longer inference time, our model does perform significantly better. We believe that the additional time could be acceptable in certain cases where the performance is much more important than inference time, with limited memory storage. We do acknowledge that the increased inference time is an important limitation of our SPS method and will definitely aim to reduce this in our future work.

---

> > ### Author Response · Authors · 2020-11-24
> > **Response to reviewer #3 - general thanks and comments (2/2)**
> >
> >
> > In summary, our SPS method provides a new approach in Parameter Sharing, in which we use a shuffling mechanism to improve performance. To the best of our knowledge, this is the first approach in Parameter Sharing that uses shared internal parameters in different positions to train the parameters more efficiently. We believe our SPS is meaningful, because it has shown considerable and consistent performance improvement in a relatively simple and intuitive manner. SPS is also widely applicable to many other models that apply parameter sharing. For example, although not mentioned in the paper, we applied SPS in transformer encoder-decoder models for Neural Machine Translation tasks and found that SPS improves the performance by an average of 0.5 BLUE scores in IWSLT’14 datasets. We believe that further studies on shuffling mechanism can derive a number of invaluable subsequent studies, such as designing more parameter-efficient models by determining which parameters are interchangable or distinct on their own.
> >
> > we have also included more ablation studies to prove the effectiveness of PTP (first paragraph in Section 4.4) and additional explanation behind how we came up with our approach and why we intuitively thought this will improve performance (last paragraphs in Sections 3.2 and 3.3). We hope these revisions made our paper more robust. We would truly appreciate it if you could review our updated paper once more. Thank you.

---

### Official Review · AnonReviewer1 · 2020-10-30
**A Method to Expand and Initialize Student Model of Knowledge Distillation**

**Rating:** 7
**Confidence:** 3

**Review:**

This paper proposed a framework for knowledge distillation with smaller number of parameters.The authors proposed a new parameter sharing method that allows a greater model complexity for the student model. Another contribution is that a KD-specialized initialization method named Pretraining with Teacher’s Predictions can improve the student's performance. The author combined these two methods to improve the performance of the student model on existing tasks, which has surpassed the existing knowledge distillation baseline.

The SPS method is a very natural idea of extending the student model. It expands the model complexity of the student model and improves the representation ability of the model without increasing the number of parameters. The idea of PTP is an excellent initializing method. At the same time, the teacher model's generalization of knowledge is given to students through initialization, and then fine-tuned.

The experimental part of this article is also quite sufficient. The ablation experiment illustrates the effectiveness of the two steps for the optimization of results.

---

> ### Author Response · Authors · 2020-11-24
> **Response to reviewer #1 - general thanks and comments**
>
> We would like to thank Reviewer 1 for the detailed and thoughtful review. We were delighted to hear that you found our core ideas interesting. Based on reviewers' feedback, we have included more ablation studies to prove the effectiveness of our method (Section 4.4) and additional explanation behind how we came up with our approach and why we intuitively thought this will improve performance (Sections 3.2 and 3.3). We hope these revisions made our paper more robust. We would truly appreciate it if you could review our updated paper once more. Thank you.

---

### Decision · Program_Chairs · 2021-01-07
**Final Decision**

**Decision:**

Reject

**Comment:**

This paper got mixed reviews. One for acceptance, one for reject and two borderline. After the rebuttal, AR2 raises the review to borderline.  AR1 gives the highest recommendation but did not provide detailed supporting evidence. Other reviews provide comment on the strength and also share the concerns. Most of the concerns concentrate on the motivation (whether the proposed method is violating the objective of knowledge distillation) and the brought additional computation overhead. Also the scope of this paper was defined wider than the actual one. The authors only did experiments for BERT but did not consider and compare with existing KD method. Overall, AC read the paper and also has the similar concerns, the novelty is limited. the brought increase in inference time is violating the KD objective and the scope of this paper was not defined clearly. The authors should improve the submission in these aspects. At its current status, AC does not recommend acceptance.